# Resolution limit of the eye – how many pixels can we see?

Maliha Ashraf [1], Alexandre Chapiro [2] & Rafał K. Mantiuk [1,2] ✉

As large engineering efforts go towards improving the resolution of mobile, augmented reality (AR), and virtual reality (VR) displays, it is important to know the maximum resolution at which further improvements bring no noticeable benefit. This limit is often referred to as the retinal resolution, although the limiting factor may not necessarily be attributed to the retina. To determine the ultimate resolution at which an image appears sharp to our eyes with no perceivable blur, we created an experimental setup with a sliding display, which allows for continuous control of the resolution. The lack of such control was the main limitation of the previous studies. We measure achromatic (black-white) and chromatic (red-green and yellow-violet) resolution limits for foveal vision, and at two eccentricities (10° and 20°). Our results demonstrate that the resolution limit is higher than what was previously believed, reaching 94 pixels per degree (ppd) for foveal achromatic vision, 89 ppd for red-green patterns, and 53 ppd for yellow-violet patterns. We also observe a much larger drop in the resolution limit for chromatic patterns (red-green and yellow-violet) than for achromatic patterns. Our results provide quantitative benchmarks for display development, with implications for future imaging, rendering, and video coding technologies.

As display resolutions approach the limits of human perception, often known colloquially as the "retinal resolution" or "eye-limiting resolution", the law of diminishing returns begins to apply and further growth produces negligible improvements in perceived quality. The focus of our study is to measure this visual end-point of display resolution. In contrast, previous works on retinal or visual system resolution targeted individual mechanisms, for example, by eliminating optical aberrations[1], or focused on visual system performance in motion[2], shape[3] discrimination, or in Vernier acuity tasks[4]. Our goal is to determine the resolution limit for a high-quality display, which appears indistinguishable from a perfect reference. Beyond the resolution limit of foveal vision, our work also sheds light on important ancillary variables unexplored in previous research, such as the resolution limits in retinal eccentricity (fovea versus periphery), and for (isoluminant) colour modulations.

The resolution limit of the eye is influenced by both optical and neural factors, which interact in a complex and often unintuitive manner. When the light enters the eye, it is scattered in the ocular media[5] and imperfectly focused on the retina due to optical aberrations[6,7]. The diffraction on the pupil restricts the maximum frequency that the eye can resolve[8,9]. Because of errors in accommodation, the resolution limit can vary with the viewing distance[10–12], resulting in a higher resolution limit at larger distances. We discuss this effect in the context of our study in the Supplementary Section 4.5 ("The effect of viewing distance"). The resolution is also limited by the spacing of the photoreceptors[13] and retinal ganglion cells[14]. Cones are densely packed in the fovea and decrease in density towards the periphery[15]. Retinal ganglion cells (RGCs), which pool information from the photoreceptors, show a similar distribution across the retina[16]. Thus, depending on the stimulus colour modulation and its position on the retina, any combination of optical and neural anatomical factors can be limiting the resolution of the eye[17–19].

The key challenge of measuring the resolution limit is to provide precise control of the resolution (spatial frequency) of the displayed

[1]Department of Computer Science and Technology, University of Cambridge, Cambridge, UK. [2]Meta, Applied Perception Science, Sunnyvale, CA, USA. ✉e-mail: rafal.mantiuk@cl.cam.ac.uk

stimulus. A stimulus shown on an electronic display can be precisely reproduced only for the display's native resolution and its integer downsampling factors. Showing intermediate resolutions requires digital resampling, which alters the frequency content of the displayed stimulus. To avoid this problem, we employ a mechanised apparatus in which we can move the display towards or away from the observer (Fig. 1), which lets us smoothly control the displayed resolution. By doing so, we reproduce the 130-year-old study of Wertheim (1894)[20,21] who used wire gratings to determine the relative drop of the resolution limit in parafoveal vision. Modern technology lets us employ robust psychophysical techniques, provide absolute measures of resolution, and include measurements for both achromatic and chromatic patterns.

The quantitative findings of this study have wide-ranging implications across multiple fields. For consumer electronics, understanding the limits of perceptible resolution can guide the development of more efficient and cost-effective displays. In virtual and augmented reality (VR/AR/XR), the optimisation of visual content based on perceptual thresholds, particularly in terms of retinal eccentricity and chromatic modulation, can enhance user experience without unnecessary computational costs. Additionally, in video compression, evidence-based chroma subsampling schemes can improve coding performance without affecting visual quality.

## Results
### Foveal resolution limit
Figure 2a reports the measured resolution limit for achromatic, red-green, and yellow-violet patterns. The experiment measured the highest resolution that can be reliably detected by an observer at various eccentricities from the fovea. The resolution is expressed in pixels per degree or ppd. The corresponding maximum spatial

frequency in cycles per visual degree is equal to half of the reported ppd values. The thresholds are reported for detecting a high-contrast Gabor patch as well as for identifying a decrease in resolution in black text on a white background, and white text on a black background simulating dark mode. The detailed breakdown of the mean, median, and 95th percentile threshold ppd values observed across different colour directions and eccentricity levels are provided in the Supplementary Table 2. First, we will focus on the results for foveal vision, corresponding to an eccentricity of 0°.

The widely accepted 20/20 vision standard, established by Snellen, suggests that the human eye can resolve detail at an angular resolution of 1 min of arc, which corresponds to 60 pixels per degree (ppd) (Supplementary Section 2–"Visual acuity units conversion") details the conversion between these units). This measure is derived from the design of the Snellen chart, where the smallest letters on the 20/20 line subtend an angle of 5 min of arc, with each critical feature of these letters subtending 1 min of arc when viewed from 20 feet or 6 m[22,23]. This 1 arc minute value has historically been considered the threshold for human visual resolution, discussed in more detail in the Supplementary Section 1 ("Historical context of visual acuity standards"), leading to the assumption that 60 ppd is sufficiently high for display purposes. However, younger observers with no optical abnormalities usually have acuities better than 20/20. In the context of displays, the Ultra Retina XDR display found in the 7th generation Apple iPad Pro (2024, 13") has an effective resolution of 65 ppd when viewed from 35 cm away, the shortest comfortable viewing distance. Both the 20/20 assumption and the Retinal display resolutions are significantly lower than the population mean of 94 ppd we measured in our experiment, or individual values as high as 120 ppd (see Supplementary Fig. 5). This demonstrates that the 60–65 ppd range is not the "retinal resolution" for a display. Note that high-contrast content, such

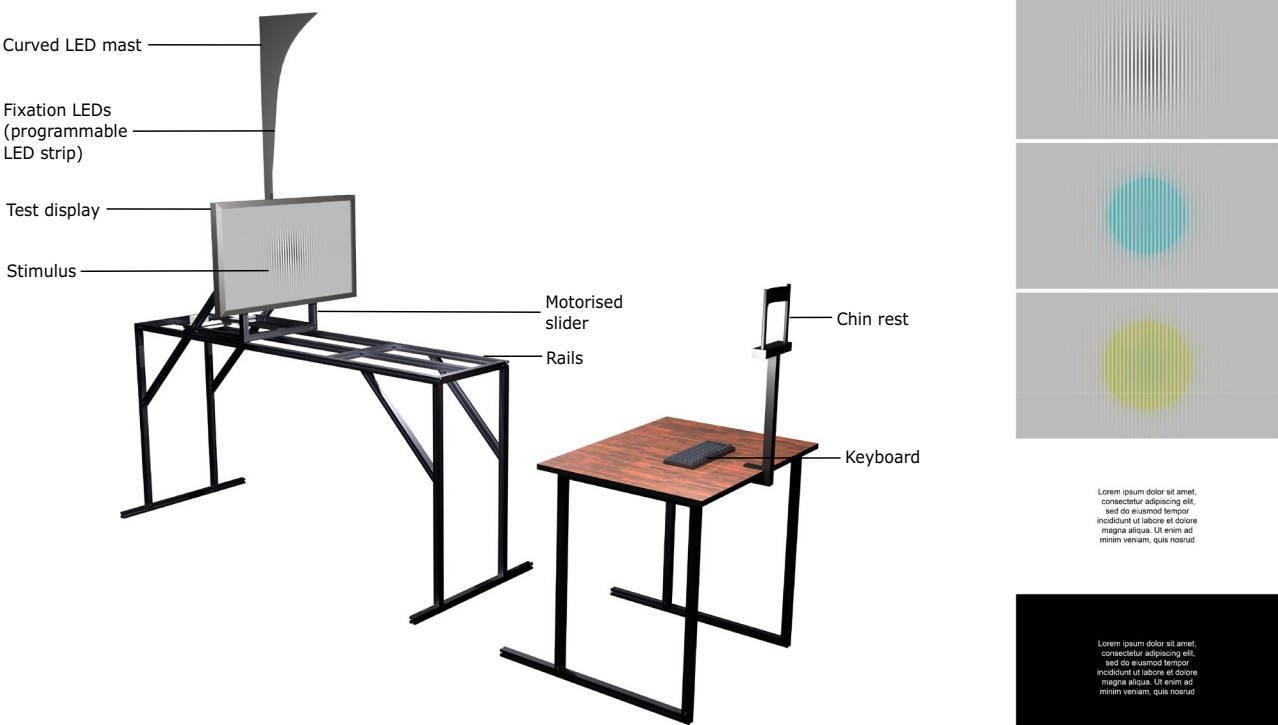

**Fig. 1 | Experimental setup.** Left: Rendition of the experimental setup. The display can slide on rails towards and away from the observer. The movement is controlled by a motorised camera slider to present stimuli at different pixel-per-degree (ppd) resolutions. The fixation point for the foveal presentation is the black cross in the centre of the screen. For peripheral viewing, an LED on the curved LED mast is lit up, corresponding to the retinal eccentricity. The curvature was designed to approximate the distance to the horopter (for the average display position). The photograph of the actual apparatus can be found in Supplementary Fig. 1. Right: Stimuli used in the experiments. From top-to-bottom: achromatic, red-green and yellow-violet square-wave gratings, black-on-white text, and white-on-black text.

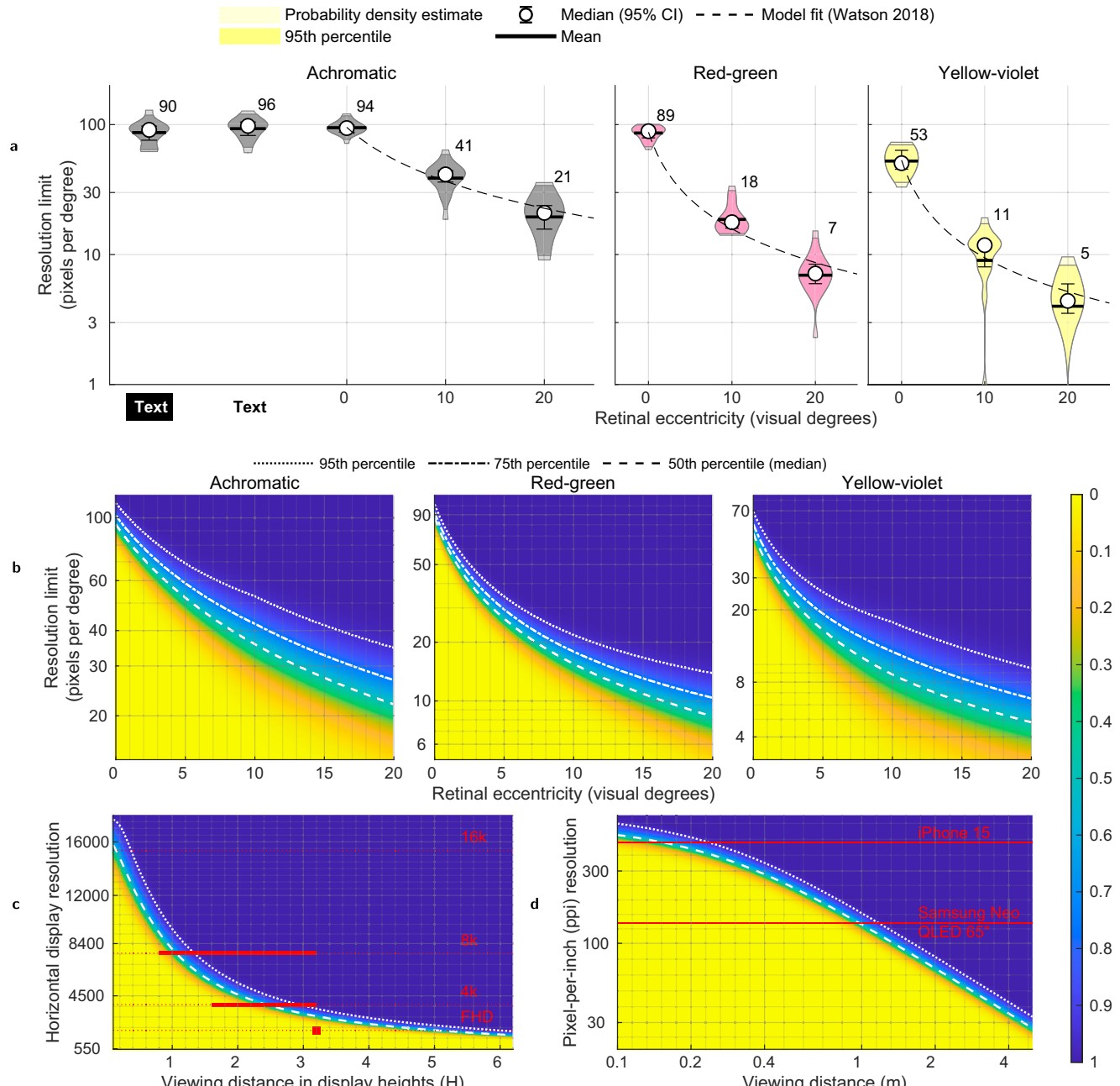

**Fig. 2 | Spatial sensitivity and resolution limits for various colour directions across the visual field. a** The measured resolution limit in pixels-per-degree (ppd) at each eccentricity across the sample (N = 18), with median (open circles), 95% Confidence Intervals (CIs; error bars), and mean (horizontal bars). Numbers next to the violins indicate median ppd values of the observed data. Dashed lines represent the model fit. The edges of the shaded violin plot areas indicate the 95th percentile of thresholds. **b** Heatmap showing the cumulative probability density of resolution limits within the observer sample, centred around predictions from the fitted Watson (2018)[30] model. The model was fitted using measured resolution limits at specific eccentricities (Fig. 2a) and interpolated across those eccentricities to provide a continuous representation. **c** Ideal display vertical resolution as a function of viewing distance expressed in display height (H). The red horizontal bars indicate the ITU-R BT.2100-2[31] recommended viewing distances for various display resolutions: FHD (2K), 4K, and 8K. **d** Required pixel-per-inch (ppi) resolution needed as a function of viewing distance (meters). In plots (**b**–**d**), blue areas indicate that any further increase in pixel resolution would not be perceptible to almost all observers, while yellow areas represent resolutions that will be within the visual perceptual limit of almost all observers. The dotted and dashed lines represent different percentiles of the sample as shown in the legend.

as our Gabor patch stimulus, is not outside of the norm for content typically seen on displays: notably, text is often rendered at maximum contrast. To demonstrate, we also measured the detection threshold for text, both black-on-white and white-on-black (dark mode), and obtained values closely matching the resolution limit for sinusoidal gratings, as indicated by the "Text" data points in Fig. 2a. Our results

clearly indicate that the resolution limit of the eye is higher than broadly assumed in the industry.

It could also be surprising that the foveal resolution limit of red-green patterns is similar to that of achromatic patterns—89 ppd for red-green vs. 94 ppd for achromatic. It must be noted, however, that we did not try to isolate observers' individual chromatic mechanisms

via the heterochromatic flicker paradigm[24], as we wanted to capture data that could generalise across the population. Our results cast doubt on the common practice of chroma sub-sampling found in almost every lossy image and video format, from JPEG image coding to H.265 or AV1 video encoding. The assumption of chroma subsampling is that the resolution of chromatic channels can be reduced twofold in relation to the achromatic channel due to the lower sensitivity of the visual system to high-frequency chromatic contrast. Our data suggests that this only holds for the yellow-violet colour direction, with the maximum resolution of 53 ppd, but not for the red-green direction, consistent with the vision science theory that the isoluminant red-green pathway is the most sensitive opponent-colour channel of the human visual system[25].

## Resolution limit in periphery

Figure 2a shows the rapid decline of the resolution limit as the stimulus was presented at increased eccentricities. This is in line with the established understanding that visual acuity and colour discrimination decrease as the stimulus moves away from the fovea, primarily due to the fall-off in cone density and the increase in receptive field size in the retina[4,26].

The notable aspect of our results is that the resolution limit declines with increased eccentricity differently across colour directions. The achromatic resolution limit declines 2.3× between foveal vision and 10° eccentricity, while red-green declines 4.9× and yellow-violet 4.8×. Popular techniques, such as foveated rendering[27,28] or foveated compression[29], are optimised for achromatic vision. Our results suggest that these techniques could provide further computational and bandwidth savings by lowering the resolution requirements for the chromatic channels.

## Modelling the resolution limit

To interpolate and extrapolate our measurements, we fit the contrast sensitivity model presented by Watson (2018)[30]:

$$\log(S^c(e, \rho)) = \log(S_0^c) + k_\rho^c(1 + k_e^c e)\rho, \quad \forall c \in \{\text{Ach}, \text{RG}, \text{YV}\}, \quad (1)$$

where $S^c$ is the contrast sensitivity of the colour channel $c$ for a given stimulus at eccentricity ($e$) and spatial frequency ($\rho$). $S_0$ is the baseline sensitivity affected by other stimulus parameters (luminance, temporal frequency, size, etc). $k_\rho$ and $k_e$ are the parameters of the model representing linear decrease with respect to spatial frequency and retinal eccentricity, respectively. The contrast sensitivity $S$ is the inverse of the contrast of the stimulus. In our study, the contrast value is fixed for each colour direction (values reported in the Supplementary Table 1). We optimise the values of $S_0$, $k_\rho$ and $k_e$ to predict the measured $\rho$ values from our data. The rearranged equation predicting the spatial frequency threshold as an inverse factor of eccentricity follows:

$$\rho(e) = \log\left(\frac{S^c}{S_0^c}\right)\frac{1}{k_\rho^c(1 + k_e^c e)}, \quad \forall c \in \{\text{Ach}, \text{RG}, \text{YV}\}. \quad (2)$$

The fitted model is drawn as dashed lines in Fig. 2a. More details of the fitting procedure and the parameter values are provided in the Supplementary Section 4.3 ("Parameters of the resolution limit model").

## Resolution limit across the population

In practical applications, it is important to know how the resolution limit varies across the population. This lets us make decisions that are relevant for the majority of the population. For example, designing a display which has "retinal resolution" for 95% of people rather than an average observer. To model the variation of the resolution limit in populations, we used the model from Eq. (2) to find the mean threshold, and then fitted a normal distribution to the per-observer

data. To estimate the probability distribution at eccentricities not measured in our dataset, we linearly interpolated the parameters of the Gaussian distribution as detailed in the Supplementary Section 4.4 ("Probability distribution across the population"). The cumulative distribution, shown in Fig. 2b, demonstrates a large variation across the population, especially at eccentricity. For example, if a median observer can see up to 22 ppd at 20° eccentricity, this value increases to 35 ppd for the 95th percentile of our sample. This shows the importance of considering individual differences in populations when designing technology aligned with human vision. Additionally, we also tested the effect of viewing distance on the resolution limit, but did not observe a consistent trend among our sample of observers. More details of this investigation are discussed in the Supplementary Section 4.5 ("The effect of viewing distance").

We may also want to know how the resolution limit, expressed in ppd units, translates to actual displays and viewing distances. This is shown in Fig. 2c, where we plot the relationship between the display resolution (number of horizontal lines) and the viewing distance (measured in display heights). Our model predictions can be compared with the ITU-R BT.2100-2[31] recommended viewing distances for television, shown as red horizontal lines in Fig. 2c. Since Full HD (FHD) resolution was not designed to deliver a perfect image, the ITU recommendation of 3.2 display heights falls short of the reproduction below the visibility threshold. Our model indicates that a distance of at least 6 display heights would be necessary to satisfy the acuity limits of 95% of the observers. For 4K and 8K displays, the ITU suggests viewing distances of 1.6–3.2 and 0.8–3.2 display heights, respectively. Our model shows that those ranges are overly conservative and there is little benefit of 8K resolution when sited further than 1.3 display heights from the screen. Used in this way, our model provides a framework to update existing guidelines and to establish new recommendations based on the limitations of our vision. In Fig. 2d, we plot the relation between pixel density (in pixels-per-inch) and viewing distance and show the screen resolution for two different devices. To allow the readers to test their own displays, we created an online display resolution calculator available here.

## Example: foveated rendering

Foveated rendering, found in many commercial XR headsets, reduces the quality of rendered content depending on how far is a portion of the screen from the gaze location[27,28]. Foveated rendering typically reduces the resolution of rendered content to save bandwidth and the computational cost of rendering. The majority of foveated rendering methods consider only the perception of achromatic contrast and are manually tuned. Here, we show how our measurements could be used to find the right thresholds for foveated rendering for achromatic and chromatic contrast.

Here, we consider a simplified task of foveated filtering, which could improve video compression in foveated streaming—we remove the high-frequency contrast that is invisible to the human eye. To produce the result shown in Fig. 3, we decompose an image into achromatic, red-green and yellow-violet components of the DKL colour space[32], decompose it into frequency bands using a Laplacian pyramid[33], and then set to zero the coefficients below the threshold contrast for a given eccentricity according to our fitted model for an average observer. More details of the image simulation can be found in the Supplementary Section 4.6 ("Foveated filtering"). When the reconstructed image in Fig. 3 is seen from the right distance, and the gaze is directed towards the 0° target, the loss of resolution at larger eccentricities should be invisible.

## Discussion

As the aim of our experiments was to determine the ultimate resolution of a display, our methods and results differ from those found in studies on the resolution limit of the visual system. Here, we briefly

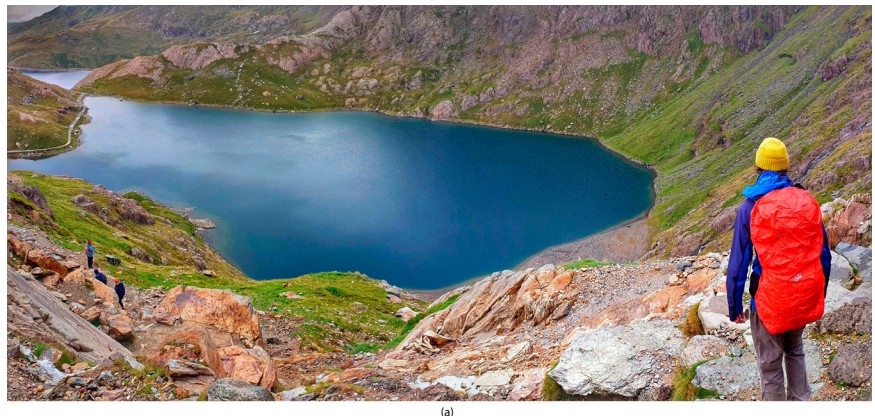

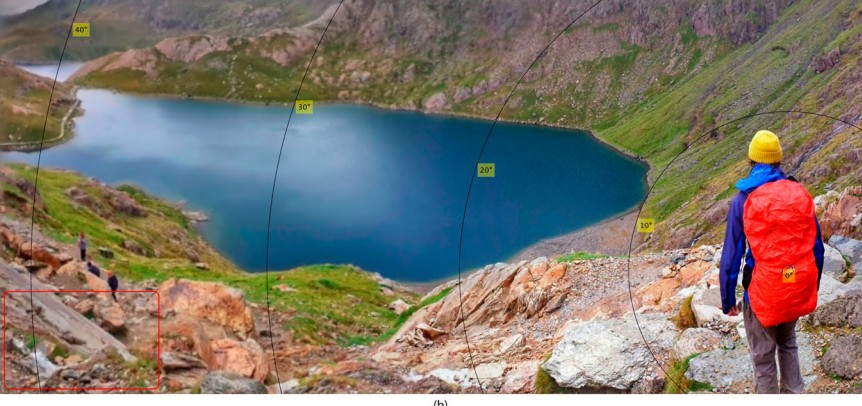

**Fig. 3 | Eccentricity-dependent filtering that removes invisible details to improve coding performance. a** Original image. **b** Filtered image; the contour lines show the retinal eccentricity positions relative to the gaze position. The filter was applied uniformly across discrete segments of eccentricity to better show the differences. Please zoom the page on the screen such that the red rectangle in the bottom-left corner of the simulated image is approximately the size of a credit card and view the image from 50 cm away. When the gaze is centred on the red backpack in the image, the degradation of high-frequency details in the periphery will not be noticeable to the human eye.

discuss the most relevant studies, listed in Table 1. We focus on psychophysical measurements that consider all stages of vision, but we do not review physiological measurements of individual mechanisms (e.g., retinal ganglion cell receptive field sizes[34] or cone distribution and size in peripheral vision[35]).

Our methodology precisely controls the pixel-per-degree resolution incident on the visual system by changing the viewing distance, as well as resampling images by integer factors. This lets us explore the visual system's response to different spatial frequencies under controlled conditions. This approach is reminiscent of Wertheim's study from 1894[20,21], which highlighted differences in visual acuity across the visual field, but only reported the relative measurements of visual acuity. Weymouth et al. (1928)[36] reported carefully measured visual acuities with variable frequency gratings but only for the central 1.5° field of view. Others have utilised context-rich stimuli like Landolt rings[3], letters[26,37], or images[38]. Such measurements explain the resolution needed to achieve specific discrimination performance (e.g., determine the orientation of "c"), but these results do not directly translate into image quality.

Some studies attempted to isolate optical and neural factors and used specialised techniques such as interference fringes to project stimuli directly onto the retina, bypassing the eye's optics[1], or used Maxwellian systems to converge light onto the pupil[39]. These methodologies can help us understand which part of the visual system is the bottleneck, but they do not account for the everyday viewing experiences where the eye's optics play an important role. Our method addresses this by assessing the visual system's response as a whole, considering both the optics of the eye and the processing

capabilities of the retina and higher neural mechanisms, and the resolution limits we report should be interpreted as behavioural thresholds. This approach is relevant for applications such as display design, where the practical concern is whether a stimulus is detectable by an observer under naturalistic viewing conditions. Thus, while we acknowledge that neural factors, such as retinal ganglion cell sampling or cortical receptive field properties, may constrain spatial resolution, our study does not attempt to identify the relative contributions of these mechanisms.

A distinction should also be made between detection and discrimination measurements. For example, Anderson et al. (1991)[2] used a direction discrimination paradigm, and Healy & Sawant (2012)[40] reported discrimination thresholds for different shapes, colour, and size features. Detection thresholds for simple stimuli, which only involve the observer perceiving the stimulus, are often higher than discrimination thresholds, where the observer must not only perceive but also identify and discriminate the stimulus (in terms of orientation, motion direction, etc.). Anderson et al. (1991)[2] also eliminated artefacts induced by high spatial frequency aliasing, so their measurements do not include stimuli that can still be detected but are perceived as distorted or aliased artefacts.

An important consideration in interpreting our results is the role of binocular versus monocular vision. Our study measured binocular acuity, which better represents real-world viewing conditions where both eyes are used. Binocular acuity is generally higher than monocular acuity[41–43] due to binocular summation[42,43] and reduced optical aberrations when both eyes contribute to perception[41]. Measuring resolution thresholds for nasal and temporal fields in binocular

**Table 1 | Visual acuity measurements from the literature**

| Study | Eccentricity range | Colour modulation | Diameter (°) | Data reported as | Stimuli and methodology |
|---|---|---|---|---|---|
| Wertheim (1894)[20,21] | 0°–70° | Achromatic | 1°–3° | Relative visual acuity (temporal) for 1 observer | Gratings made of high-precision wire frames[51] mounted on a moving rig viewed monocularly |
| Weymouth et al. (1928)[36] | 0°–85° | Achromatic | 2.13° | Minimum angle of resolution (MAR) averaged across all meridians for 1 observer; values reported in Weymouth (1958)[3] | Square-wave interference fringe (Ives visual acuity object) with variable spatial frequency viewed monocularly |
| Ludvigh (1941)[37] | 0°–10° | Achromatic | — | Snellen fraction averaged for 3 observers | Individual Snellen letters viewed monocularly |
| Weymouth (1958)[3] | 0°–20° | Achromatic | — | MAR averaged for 20 observers | Landolt C ring viewed monocularly |
| Kerr (1971)[39] | 0°–30° | Achromatic | 3° (square) | Log visual acuity values averaged for 2 observers | Square-wave grating viewed monocularly via a Maxwellian apparatus |
| Anstis (1974)[26] | 4°–55° | Achromatic | — | Threshold letter heights in visual degrees averaged for 2 observers | Letters in a custom radial chart viewed binocularly |
| Thibos et al. (1987)[1] | 0°–35° | Achromatic | 3°, 2.5° | Spatial resolution (cpd) | Sinusoidal interference fringe directly on the retina viewed monocularly |
| Anderson et al. (1991)[2] | 0°–55° | Achromatic, red-green | 0.23°–7.5° | Spatial resolution (cpd) | Sinusoidal gratings viewed monocularly in motion-discrimination and detection tasks |
| Masaoka et al. (2013)[38] | Free viewing | — | 3.2° (square) | Spatial resolution (cpd) | Colour images displayed at different resolutions and compared with real objects |

Summary of classical and modern studies of eccentricity-dependent visual acuity. "cpd" denotes cycles per degree. Please refer to Supplementary Section 2 ("Visual acuity units conversion") for conversion between different units of visual acuity.

vision introduces a complexity: a stimulus presented at a given eccentricity on the nasal side of one eye will appear at a lower eccentricity on the temporal side of the other eye. This does not happen when testing superior and inferior visual fields, which is what was used in our study.

The resolution limits reported here capture the most conservative resolution limit—the point at which even a high contrast stimulus could not be detected. Figure 4 provides a direct comparison of resolution thresholds from datasets whose methodologies were comparable with ours. Most of these studies report lower resolution limits than those measured in our study, particularly in the peripheral regions. In contrast to our controlled threshold study, Masaoka et al. (2013)[38] measured the "realness" scores for complex scenes at a fixed set of resolutions, which were produced by digital resampling. They observed that the scores began tapering off between the 53 and 78 cycles per degree (cpd) resolutions. Based on this, they estimated a resolution limit of approximately 60 cpd but did not directly measure any thresholds at this resolution. It should also be noted that natural images (such as those used in their study) rarely contain high contrast at high spatial frequencies (due to an expected 1/f power distribution)[44], though artificial content, such as text, often does contain such high contrasts.

The use of square-wave gratings in our study introduces higher harmonic frequencies, which may contribute to higher contrast sensitivity limits compared to studies using sinusoidal gratings. Campbell & Robson (1968)[45] demonstrated that at high spatial frequencies, the contrast sensitivity curves for sinusoidal and square-wave gratings exhibit similar shapes due to the visual system's inability to resolve higher harmonics, which have higher frequency and lower amplitude than the fundamental frequency. Effectively, at higher spatial frequencies, we are detecting the fundamental frequency for both types of gratings. The amplitude of the fundamental frequency component of the square-wave is $4/\pi$ (1.273) higher than that of the sinusoidal wave of the same contrast[45]. Therefore, if we were to replace square waves with sine waves, the contrast of the corresponding sine waves would be $4/\pi$ higher.

When comparing our results with historical data, it is evident that while our methodology leverages modern technological advances, the fundamental characteristics of human visual acuity documented in earlier works remain consistent. Our study reaffirms the findings of earlier investigations into retinal resolution but also expands the scope by employing a holistic approach that considers the entire visual system. This approach provides a more accurate representation of visual performance in real-world viewing situations, offering valuable insights for the design of displays and other applications where visual clarity and resolution are critical.

## Methods

Below we summarise the main points of the methods, while the detailed description can be found in the supplementary.

### Moving display apparatus

We designed an experimental apparatus capable of adjusting the display's position relative to the observer to smoothly vary the effective resolution (in terms of pixels per visual degree) without the need for pixel resampling. The apparatus features a 27-inch Eizo ColorEdge (CS2740) 4K monitor mounted on a mobile cage, allowing it to move smoothly along a 1.6-m track as shown in Fig. 1. This setup enables a wide range of viewing distances. The movement is controlled by a stepper motor via an Arduino Uno microcontroller. Central fixation for foveal stimuli is provided by a black cross at the centre of the display. For stimuli at larger eccentricities, the fixation point is adjusted using specific LEDs on a programmable LED strip controlled by a Raspberry Pi Pico microcontroller.

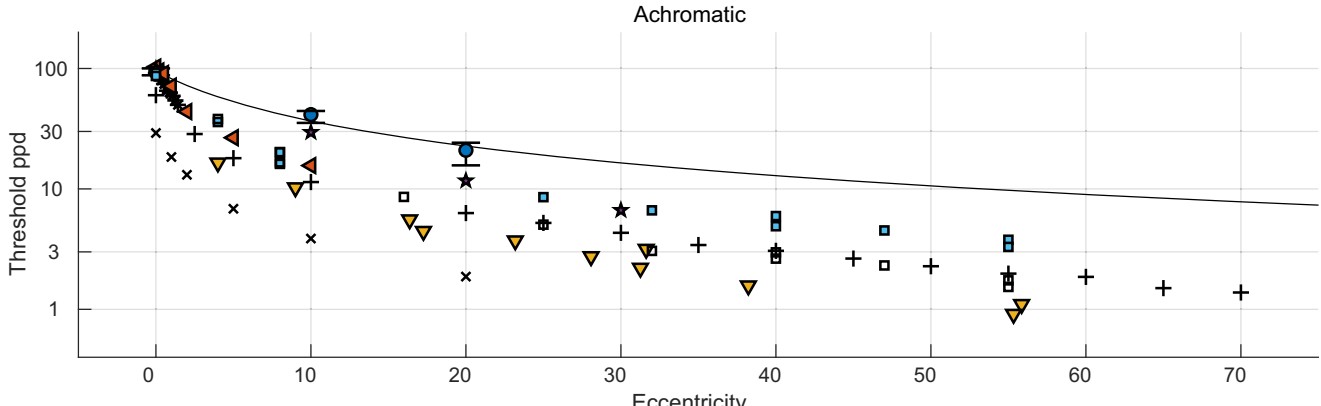

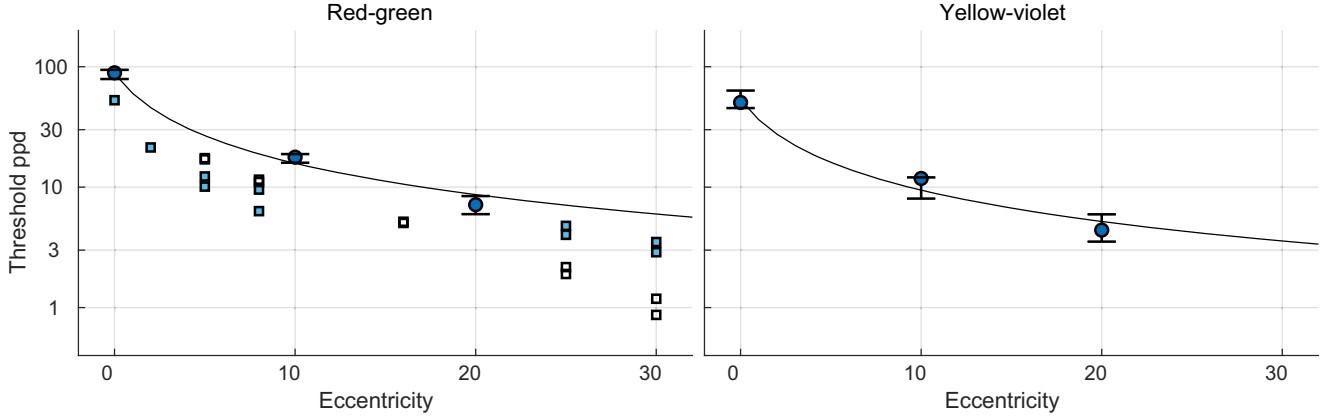

**Fig. 4 | Comparison of resolution thresholds with previous models and datasets.** Resolution limits from the present study (median values, error bars denote 95% confidence intervals; $n = 18$ observers) are plotted against eccentricity and contrasted with thresholds reported in earlier studies for achromatic, red-green, and yellow-violet stimuli. The Watson (2018)[30] model fit is shown for reference (solid line). Most previous datasets report lower resolution limits than those measured in our work.

## Stimuli

For the gratings experiments, the stimuli consisted of square-wave gratings, modulated with a Gaussian envelope, with an underlying frequency equal to the Nyquist frequency of the display. These stimuli are widely used in vision science experiments because the foundational visual detectors of the human visual system are likely optimised for similar waveforms[46]. The gratings were modulated along three directions in the DKL colour space[32]: achromatic (L+M) with a contrast of 0.96, chromatic (L−M) with a contrast of 0.23, and chromatic (S-(L+M)) with a contrast of 0.89. The contrasts were chosen to ensure they could be faithfully reproduced on the display. The resolution (spatial frequency) of the stimuli was modulated by moving the display towards or away from the observer and by upsampling or downsampling the spatial frequency of the Gabor patch using integer factors of 2, 3 and 4. The standard deviation of the Gaussian envelope of the square-wave gratings was adjusted to always subtend 2 visual degrees, to ensure a consistent stimulus size across different viewing distances and eccentricities. The stimuli were shown at three eccentricities: 0°, 1° and 20°, and at luminance level 100 cd/m².

A different procedure was used for the experiment involving text stimuli. We created two images with the same text but in opposite polarities; one with a white background and black text, and the other with a black background and white text. The contrast between the text and the background in both images was 0.96. The task was to detect a sharper stimulus from the two: one rendered at the native resolution of the display and one upscaled from the lower resolution using a $n \times n$ box filter. The reported ppd values correspond to the effective resolution of the upscaled image (reduced by the factor of $n$). To ensure that the text was the same size regardless of viewing distance, the native resolution image was obtained by resampling a high-resolution reference image using a Lanczos filter ($a = 3$), so that the stimulus spanned 12.63° × 7.12° visual degrees at all viewing distances. For reference, the height of a capital letter (e.g., "I") was 0.28° and the body size (i.e., the full vertical extent of the text, including any parts that extend above or below the typical height of capital letters) was 0.5°. At a typical reading distance of 40 cm, this corresponds to a capital letter height of 5.6 pt, and a font size of 10 pt in standard printed documents[47]. For the text experiment, the luminance of the white background was 191 cd/m², and for the black background, it was 8.67 cd/m². Only foveal stimuli with free viewing were used for the text stimuli. The stimuli used in our experiments are shown in Fig. 1. More details of the stimuli generation methods are provided in the Supplementary Section 3.2 ("Stimuli").

## Psychophysical method

We used a two-interval-forced-choice (2IFC) paradigm. Participants were seated in a dark room, facing the display such that the viewing distance was between 110 and 270 cm depending on the values of the tested ppd for the stimuli. Each interval was preceded and ended with the presentation of a masker with a random noise to minimise the chance that the temporal change could aid detection. Participants indicated which interval contained the stimulus, and their responses were recorded. Each of the trials was repeated 3 times consecutively as readjusting the display position after each trial would extend the duration of the experiment. The QUEST[48] adaptive procedure was used to select the next pixel-per-degree resolution to be tested based on participants' responses. The trails for achromatic, red-green and yellow-violet colour directions were interleaved (in random order), which required maintaining 3 separate QUEST states. The threshold for each observer was estimated from 30 to 50 QUEST trials. The trials stopped when either the maximum allowed number of trials was reached or the standard deviation of the threshold estimate determined by `QuestSd` function from Psychtoolbox[49] reached a value of 0.07. The value of the stopping criterion was chosen to ensure sufficient precision in the threshold estimate while keeping the experiment duration manageable and was determined based on initial testing with a small group of observers.

## Observers

Eighteen observers (6 female, 12 male) with a mean age of 25.5 (age range: 13–46 years) participated in the first part of the experiment (gratings stimuli). For the second part of the study (text stimuli), twelve of the original participants (3 female, 9 male) were involved. All had normal or corrected-to-normal vision and normal colour vision, tested with Ishihara colour plates. Visual acuities were tested using Snellen charts (more details in the Supplementary Section 4.2 – "Relationship with visual acuity"). The experiment was approved by the departmental ethical committee at the Department of Computer Science and Technology of the University of Cambridge. Participants were recruited through university mailing lists, and most were students or members of the lab group. Before participation, all observers were briefed on the purpose of the research, the procedures involved and their rights as participants, including the right to withdraw at any time without penalty. Each participant read and signed a standard consent form outlining these rights before the experiment. The experiment was also demonstrated to the participants and they were asked to do a short trial run to ensure they understood the task before the main experiment started. All the data was anonymised and the participants were compensated for their time.

## Data analysis

We used the Maximum Likelihood Estimation method to fit psychometric functions to individual participants' binary responses from the QUEST trials. The likelihood function quantified how well a psychometric function explained the observed responses across different stimulus intensities. Outliers, identified based on the modified Z-score, were excluded from the dataset to prevent skewed estimates of psychophysical trends. In total, 13 out of 162 data points were identified as outliers and excluded from subsequent analyses. The post-hoc analysis showed that those were most likely caused by lapses of attention. The individual data points, including the outliers, are shown in the Supplementary Fig. 5.

## Reporting summary

Further information on research design is available in the Nature Portfolio Reporting Summary linked to this article.

## Data availability

The raw and processed data generated in this study have been deposited in the GitHub repository https://github.com/gfxdisp/resolution_limit and archived[50].

## Code availability

The code for reproducing the plots from Fig. 2 can be found at https://github.com/gfxdisp/resolution_limit/ and archived[50]. Furthermore, a web-based calculator to convert between the physical specification of the display and the effective visual resolution can be found at https://www.cl.cam.ac.uk/research/rainbow/projects/display_calc/.

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

## Acknowledgements

We would like to extend our gratitude to Thomas Bytheway for his exceptional work in building the display rig. We also thank Scott Daly, Trisha Lian, and Minjung Kim for their feedback and suggestions. The 3D schematic of the setup (Fig. 1) was prepared by Karolina Mantiuk. This project was funded by a research grant from Meta.

## Author contributions

A.C. proposed the project. R.K.M. and A.C. conceived the experiment. M.A. designed the psychophysical study and conducted the experiment under R.K.M.'s supervision. M.A. analysed the results and created the figures with feedback from all authors. All authors contributed to writing and reviewing the manuscript.

## Competing interests

The authors declare no competing interests.
