## [Transparent Peer Review file · Nature Communications]

Resolution limit of the eye — how many pixels can we see?

Corresponding Author: Professor Rafał Mantik

Version 0:

Reviewer comments:

Reviewer #1

(Remarks to the Author)

The authors investigate the resolution limit of human eye using psychophysics experiment to measure the achromatic and chromatic contrast sensitivity for foveal and peripheral vision. The experiment is well-designed showing important findings that can guide the display development of flat-panel and near-eye displays. I recommend the authors addressing the following comments:

1. This study presents higher contrast sensitivity limits than previous studies. The Gabor targets use square wave instead of sinusoidal grating patterns. Please comment if this can affect the contrast sensitivity measurement results.
2. The angular dimension of the target may vary with the display distance. Please clarify the impact of varying target angular dimension on the measurement results.
3. The fixation point moves on an arc that adjust the eccentricity in the longitudinal direction. However, it is unclear if the result can be applied to different eccentricities in the horizontal direction with nasal or temporal eye rotation for binocular vision. Please comment in the manuscript and/or provide additional measurements.

Reviewer #2

(Remarks to the Author)

The paper "Resolution limit of the eye: how many pixels can we see?" explores the limits of human visual resolution, focusing on how these limits change based on color and eccentricity. The study's data aligns well with previous literature regarding general visual resolution capabilities. The study highlights that deviations in resolution limits are more pronounced with increasing distance from the fovea, with a max testing here of 20°, emphasizing the difference between achromatic and chromatic vision in peripheral perception. These findings provide a clearer understanding of how visual resolution varies across the visual field for different colors, particularly relevant to AR/VR designs.

Overall, I find the authors data aligns with previous literature. However, I have several concerns regarding the conclusions of their findings which are outlined below.

- Regarding how vision changes with eccentricity, at no point is the factor of eye movements mentioned. There is sufficient evidence that shows how eye movements play an important role in visual acuity and resolution limits, including saccades, microsaccades and ocular drift. This is not only missing from the introduction, but they also do not mention controlling for it in the methods. The variations in threshold ppd shown in Figure 3 compared to previous studies could be caused simply to the fact that individuals were not maintaining fixation appropriately when testing more peripheral eccentricities.
- The authors present the sliding display method as a novel technique to test different spatial frequencies; this technique however, has been used by vision scientists for decades to work around monitor resolution limits – countless times between 1894 as they suggest. Building on that, more complex system (ie. Badal system) have also been used to slide the display back and forth to test different pixel to degree conversion factors.
- In Figure 1, panel B is potentially misleading as it implies a continuous relationship between 0 and 20° eccentricities with a single data point between them. The visual presentation suggests greater resolution continuity than what is actually tested.
- The study cites Anderson (1991), which documents significant differences between nasal and temporal thresholds.

However, it is unclear whether the current study is examining these distinctions, making it challenging to compare findings directly.

-The size of the sinusoidal Gabor stimuli ($12.63^\circ \times 7.12^\circ$) exceeds the eccentricity being tested at 10° . Additional details on the Gabor's spatial envelope would provide clarity on the specific regions of the retina being stimulated. Given the dimensions of the stimuli, it is likely that even at lower contrasts, the central retina is being activated, extending beyond the 10° eccentricity to potentially cover a range from approximately 4° to 16° . This raises concerns about the validity of attributing the findings exclusively to the 10° retinal region – even when ignoring eye movements.

- The method lacks clarity regarding the QUEST protocol—whether data were gathered across 30-50 individual blocks per participant or in a single block with multiple trials. Given QUEST's known biases, multiple blocks are typically recommended to ensure more reliable threshold estimates.

- The study does not specify whether multiple monitors were used or how the display was calibrated (e.g., linearization). These details are critical for assessing the generalizability and reproducibility of the findings.

Reviewer #3

(Remarks to the Author)

This paper presents a study investigating the resolution limit of the human eye in a principled manner, addressing a critical question for display technology: at what point do further increases in display resolution cease to provide perceptual benefits? The main contribution of the paper is the development of a display system that enables continuous control of resolution. Using this system, the authors measure achromatic and chromatic resolution limits for foveal vision and two peripheral eccentricities, finding that the resolution limit is higher than previously believed.

The paper is well-motivated and clearly written, and the implications of the results would have a wide impact on display and imaging technologies, rendering, compression, and AR/VR system designs. I would love to see this paper published in Nature Communications. However, I have a few comments that I would like the authors to address in their revision:

1. With this sliding display system, I assume that you must consider whether visual acuity is independent of viewing distance. Please clarify this assumption and discuss it in the revision.
2. Given the constraints of viewing distance and display size, I assume that these factors ultimately limited the range of eccentricities that could be measured. How would you scale up the system if one wanted to measure the resolution limit at larger eccentricities?
3. In the beginning, the authors discuss how neural factors influence the resolution limit, but I do not see how these factors were considered in the study. Please discuss this.
4. I guess the vision correction of observers would be critical for results to be reproduced. This should be further elaborated in Methods section.

Version 1:

Reviewer comments:

Reviewer #1

(Remarks to the Author)

The authors have addressed all my comments.

Reviewer #3

(Remarks to the Author)

The authors fully address my comments with additional experiments, and I recommend acceptance.

Manuscript: *Resolution limit of the eye: how many pixels can we see?*

Response to reviewers' comments

We thank the reviewers for their thoughtful and constructive feedback, and we appreciate their recognition of the significance and potential impact of our work. We are encouraged that all reviewers found the study interesting and well-motivated and appreciated its relevance to display development and emerging AR/VR technologies. Here, we provide detailed responses to each reviewer's comments and describe the revisions we made to address their concerns. The original reviewer's comments are shown in blue. The changes in the manuscript are also marked in blue.

In addition to addressing all the comments, we made the following changes:

- We added detailed information about the statistical tests used
- We improved the explanation of the text stimuli
- We added section "*Data and code availability*" with the link to the repository with the data and the code: https://github.com/gfxdisp/resolution_limit

Reviewer #1

[square wave vs. sine wave] This study presents higher contrast sensitivity limits than previous studies. The Gabor targets use square wave instead of sinusoidal grating patterns. Please comment if this can affect the contrast sensitivity measurement results.

While square-wave gratings include higher harmonic frequencies, Campbell & Robson (1968) have shown that the shape of contrast sensitivity curves for sinusoidal and square-wave gratings is the same at high spatial frequencies (only high frequencies are relevant for our experiment). This is due to the inability of the visual system to resolve higher harmonics, which have higher frequency and lower amplitude than the fundamental frequency, effectively detecting fundamental frequency components of both types of gratings.

The fundamental frequency component of the square wave is 4π (1.273) higher than that of the sinusoidal wave of the same contrast /cite{campbell1968application}. Therefore, if we were to replace

square waves with sine waves, the contrast of the corresponding sine waves would be $4/\pi$ higher (see Fig. 3 in Campbell & Robson 1968).

[size of the stimulus at different viewing distances] The angular dimension of the target may vary with the display distance. Please clarify the impact of varying the target angular dimension on the measurement results.

We controlled and adjusted the angular size of the stimuli based on the motorized display's position. The standard deviation of the Gaussian envelope of the stimuli always subtended 2 visual degrees at all distances and eccentricities. We have updated the stimuli section to clarify this point.

[nasal vs. temporal directions] The fixation point moves on an arc that adjust the eccentricity in the longitudinal direction. However, it is unclear if the result can be applied to different eccentricities in the horizontal direction with nasal or temporal eye rotation for binocular vision. Please comment in the manuscript and/or provide additional measurements.

Our study primarily focused on the eccentricities along the superior visual field to establish baseline thresholds. The thresholds for nasal, temporal, and inferior thresholds could be extrapolated relative to our measurements using the data from Anderson (1991). Please note that this experiment measures binocular acuity (reflecting real-world viewing conditions), which is typically higher than monocular acuity. Measuring nasal and temporal thresholds binocularly while controlling for eccentricity introduces complexities due to differences in retinal projection. A stimulus presented at a given eccentricity on the nasal side of one eye will fall on a lower eccentricity on the temporal side of the other eye, making direct comparisons more complex. These disparities are minimised when measuring superior/inferior visual fields. We have added this discussion point to the manuscript.

Reviewer #2

[Role of eye movements] Regarding how vision changes with eccentricity, at no point is the factor of eye movements mentioned. There is sufficient evidence that shows how eye movements play an important role in visual acuity and resolution limits, including saccades, microsaccades and ocular drift. This is not only missing from the introduction, but they also do not mention controlling for it in the methods. The variations in threshold ppd shown in Figure 3 compared to previous studies could be caused simply to

the fact that individuals were not maintaining fixation appropriately when testing more peripheral eccentricities.

Our intention was to measure the resolution limit under natural viewing, which included natural eye movements (ocular drift).

We acknowledge the reviewer's concern regarding the potential influence of eye movements on visual resolution measurements, particularly in peripheral viewing conditions. Ensuring fixation compliance over an extended experimental session is challenging. We experimented with continuous eye tracking during the study, but found the system to be highly sensitive to head and facial movements, leading to frequent calibration issues, particularly for participants wearing glasses. Maintaining stable tracking for the full hour-long experiment was infeasible. Instead, we took several steps to control the possibility of undesirable eye movement. First, participants underwent a monitored training session where the experimenter ensured they understood and could maintain fixation. During the experiment, participants were explicitly instructed to maintain their gaze on a central fixation point, and the experimenter monitored fixation compliance visually.

We also performed statistical outlier removal on the main dataset to account for potential deviations from fixation, as shown in supplementary Figure E.

Finally, in response to the reviewer's concerns, we conducted a follow-up experiment with six participants from the original study, using an eye tracker (Pupil Labs Core) to record gaze position while they performed the task for 10 minutes in the 10-degree eccentricity condition. The results showed that participants were able to maintain fixation, with no significant deviations from the fixation target during stimulus presentation. We have attached below some example screenshots from three participants in the follow-up validation study, captured via Pupil Capture gaze analysis software. In each frame, the green circle marks the participant's gaze position—consistently close to the fixation-target LED (bright spot above the display). The participants' recorded eye movements are shown in the top-left of each screenshot. The red circles over the pupil indicate successful pupil detection.

We believe that these mitigations validate the reliability of our results for peripheral conditions when taken together.

The images above show screenshots from the eye-tracking software. The white rectangle is the display showing the stimulus. The green markers denote the eye's fixation points estimated by the eye tracker, and the red points are individual gaze positions. The LED used as the fixation point is visible as a brighter spot on the LED mast above the display. The bottom two screenshots show a mismatch between the LED and the measured fixation point. Because this mismatch was consistent over several seconds, we

attribute it to the inaccuracy of the eye tracker (drift) rather than the observer looking at the wrong point. The recordings were used to ensure that the observer could maintain the fixation point during the experiment.

[Novelty of the sliding display method] - The authors present the sliding display method as a novel technique to test different spatial frequencies; this technique however, has been used by vision scientists for decades to work around monitor resolution limits – countless times between 1894 as they suggest. Building on that, more complex system (ie. Badal system) have also been used to slide the display back and forth to test different pixel to degree conversion factors.

We are not aware of any studies apart from Wertheim (1894) that use the sliding display method to measure visual resolution.

The Badal systems are extensively used to study accommodative responses by presenting stimuli at varying diopters while maintaining constant angular size. They give the advantage of maintaining angular resolution with distance and the ability to explore a large diopter range, but they require very precise calibration, head constraint (a bite bar), and are typically used monocularly or in a haploscope. All these requirements make viewing stimuli through such a system very unnatural. Our study aims to measure visual resolution in real-world scenarios.

[Figure 1B continuous plot (R2)] - In Figure 1, panel B is potentially misleading as it implies a continuous relationship between 0 and 20° eccentricities with a single data point between them. The visual presentation suggests greater resolution continuity than what is actually tested.

The heatmap presented in this panel represents the model, which is fitted using our measurements at specific eccentricities (0°, 10°, and 20°) as shown in Figure 1a. Figures 1b-d show the fitted model from Watson, 2018 that allows us to interpolate resolution thresholds across eccentricities, providing a continuous representation of the spatial resolution limits based on the experimental data. This is stated explicitly in the figure caption now.

[nasal vs. temporal directions] - The study cites Anderson (1991), which documents significant differences between nasal and temporal thresholds. However, it is unclear whether the current study is examining these distinctions, making it challenging to compare findings directly.

(the same response as for Reviewer #1 comment) Our study primarily focused on the eccentricities along the superior visual field to establish baseline thresholds. The thresholds for nasal, temporal, and inferior thresholds could be extrapolated relative to our measurements using the data from Anderson (1991). Please note that this experiment measures binocular acuity (reflecting real-world viewing conditions), which is typically higher than monocular acuity. Measuring nasal and temporal thresholds binocularly while controlling for eccentricity introduces complexities due to differences in retinal projection. A stimulus presented at a given eccentricity on the nasal side of one eye will fall on a lower eccentricity on the temporal side of the other eye, making direct comparisons more complex. These disparities are minimised when measuring superior/inferior visual fields. We have added this discussion point to the manuscript.

[size of the stimulus at different viewing distances]- The size of the sinusoidal Gabor stimuli ($12.63^\circ \times 7.12^\circ$) exceeds the eccentricity being tested at 10° . Additional details on the Gabor's spatial envelope would provide clarity on the specific regions of the retina being stimulated. Given the dimensions of the stimuli, it is likely that even at lower contrasts, the central retina is being activated, extending beyond the 10° eccentricity to potentially cover a range from approximately 4° to 16° . This raises concerns about the validity of attributing the findings exclusively to the 10° retinal region – even when ignoring eye movements.

(the same response as for Reviewer #1 comment) We controlled and adjusted the angular size of the stimuli based on the motorized display's position. The standard deviation of the Gaussian envelope of the stimuli always subtended 2 visual degrees at all distances and eccentricities. We have updated the stimuli section to clarify this point.

[Clarification of the QUEST protocol (R2)] - The method lacks clarity regarding the QUEST protocol—whether data were gathered across 30-50 individual blocks per participant or in a single block with multiple trials. Given QUEST's known biases, multiple blocks are typically recommended to ensure more reliable threshold estimates.

We have provided more details of the psychophysical experiment in Supplementary Section 3.4. Our experiment consisted of three sessions/blocks, each for a single retinal eccentricity (0° , 10° , and 20°). Within each block, we measured the thresholds for three stimuli: achromatic, red-green, and yellow-violet. The trials for the three colours were randomised to minimise bias. The QUEST state was kept separately for each stimulus, allowing for non-sequential order of the stimuli. For each stimulus, we conducted 30–50 trials until the required precision in the threshold estimate was achieved. Because the

movement and distance calibration of the display took several seconds, we collected three trials at each viewing distance. The text has been extended and clarified.

[Display calibration details] - The study does not specify whether multiple monitors were used or how the display was calibrated (e.g., linearization). These details are critical for assessing the generalizability and reproducibility of the findings.

We used only one monitor. We have added the following calibration details to the supplementary section 3.1:

The display had a 10-bit colour depth and was calibrated using the Gain-Offset-Gamma (GOG) model to ensure accurate stimulus presentation. Colour measurement patches were displayed using Psychtoolbox running in MATLAB, and their spectra, along with XYZ tristimulus values, were measured using the JETI Specbos 1211 broadband spectroradiometer. This device has a luminance measurement range of 0.2 to 150,000 cd/m². All measurements were conducted in a dark room to minimise external light interference. The measurements were used to fit GOG parameters to linearise the display response. The fitted parameters were the black level values and gamma correction for the R, G, and B channels, and the 3x3 transformation matrix between XYZ and linearised RGB colourspace.

Reviewer #3

[Viewing distance and visual acuity independence] 1. With this sliding display system, I assume that you must consider whether visual acuity is independent of viewing distance. Please clarify this assumption and discuss it in the revision.

We acknowledge that viewing distance can influence visual acuity due to various optical factors, such as diffraction, pupil size, and accommodation. We mention this briefly in the main paper in the Introduction (2nd paragraph) and discuss it extensively in *Supplementary Section 4.5: The effect of viewing distance*. We ran a separate pilot experiment to determine whether the viewing distance affects our results. In short, it does not, as the range of tested viewing distances spanned only a small range of accommodative states.

In this pilot experiment, we compared detection thresholds for foveal stimuli presented at two different viewing distances ("near" and "far") across 16 observers. Although theoretical considerations suggest that factors like reduced accommodation error at farther distances or improved depth of field at closer distances may affect acuity, our empirical results showed that differences in detection performance were small and statistically significant for only two observer–stimulus combinations (both in the yellow–violet

direction). For the majority of cases, no reliable differences were found. We concluded that within the range of viewing distances tested in our setup, the impact on resolution thresholds is minimal and inconsistent. Therefore, we consider it reasonable not to correct for this factor in the main analyses.

[Scaling the system for larger eccentricities] 2. Given the constraints of viewing distance and display size, I assume that these factors ultimately limited the range of eccentricities that could be measured. How would you scale up the system if one wanted to measure the resolution limit at larger eccentricities?

In our setup, peripheral eccentricities are achieved by positioning the fixation point vertically above the screen along an LED strip on a pole. The system could be extended to larger eccentricities by simply increasing the length of this strip (perhaps with improvements to mechanical stability). At very high eccentricities, it is possible that fixation would become uncomfortable for viewers, but this was not an issue in our study.

Another possibility for high eccentricities involves employing head-mounted displays (HMDs). Commercial HMDs today have lower resolution than what was needed for foveal measurement, but their resolution could be sufficient for far peripheral conditions, where spatial sensitivity declines. The difficulty would then lie in calibrating the device at each eccentricity while accounting for the optical distortions.

[Neural factors influencing resolution limits] 3. In the beginning, the authors discuss how neural factors influence the resolution limit, but I do not see how these factors were considered in the study. Please discuss this.

In the introduction, we mention both optical and neural factors to emphasise that spatial resolution is jointly shaped by the eye's optics and the neural encoding of visual information. However, the scope of this study is not to disentangle the specific contributions of optical versus neural mechanisms, but rather to empirically measure the contrast detection thresholds that define resolution limits across different eccentricities and colour directions. These behavioral thresholds inherently reflect the combined effect of all limiting factors in the visual system. We have clarified this point in the revised discussion to better reflect the focus and interpretive limits of our study.

[Impact of vision correction on reproducibility (R3)] 4. I guess the vision correction of observers would be critical for results to be reproduced. This should be further elaborated in the Methods section.

As stated in the Methods section, all observers had normal or corrected-to-normal vision. Observers who required optical correction wore their usual glasses or contact lenses during the experiment, and visual acuity was verified using a Snellen chart prior to the experiment. To ensure that differences in acuity did not influence the results, we tested the relationship between visual acuity (measured in logMAR units) and the measured resolution thresholds. This analysis is presented in Supplementary Section 4.2 and Supplementary Figure G. We found no significant correlation between acuity and threshold pixels-per-degree values, suggesting that individual differences in acuity (with correction) did not systematically affect our results.